# All-*Trans* Retinoic Acid-Responsive LGR6 Is Transiently Expressed during Myogenic Differentiation and Is Required for Myoblast Differentiation and Fusion

**DOI:** 10.3390/ijms24109035

**Published:** 2023-05-20

**Authors:** Tomoya Kitakaze, Rina Tatsumi, Mayu Yamaguchi, Mai Kubota, Aino Nakatsuji, Naoki Harada, Ryoichi Yamaji

**Affiliations:** 1Division of Applied Life Sciences, Graduate School of Life and Environmental Sciences, Osaka Prefecture University, Sakai 5998531, Osaka, Japan; kitakaze@omu.ac.jp (T.K.); naoki.harada@omu.ac.jp (N.H.); 2Department of Applied Biological Chemistry, Graduate School of Agriculture, Osaka Metropolitan University, Sakai 5998531, Osaka, Japan; 3Center for Research and Development of Bioresources, Osaka Metropolitan University, Sakai 5998531, Osaka, Japan

**Keywords:** differentiation, fusion, LGR6, myoblast, proteasome, retinoic acid, skeletal muscle, ubiquitin, Wnt signaling, ZNRF3

## Abstract

All-*trans* retinoic acid (ATRA) promotes myoblast differentiation into myotubes. Leucine-rich repeat-containing G-protein-coupled receptor 6 (LGR6) is a candidate ATRA-responsive gene; however, its role in skeletal muscles remains unclear. Here, we demonstrated that during the differentiation of murine C2C12 myoblasts into myotubes, *Lgr6* mRNA expression transiently increased before the increase in the expression of the mRNAs encoding myogenic regulatory factors, such as myogenin, myomaker, and myomerger. The loss of LGR6 decreased the differentiation and fusion indices. The exogenous expression of LGR6 up to 3 and 24 h after the induction of differentiation increased and decreased the mRNA levels of *myogenin*, *myomaker*, and *myomerger*, respectively. *Lgr6* mRNA was transiently expressed after myogenic differentiation in the presence of a retinoic acid receptor α (RARα) agonist and an RARγ agonist in addition to ATRA, but not in the absence of ATRA. Furthermore, a proteasome inhibitor or *Znrf3* knockdown increased exogenous LGR6 expression. The loss of LGR6 attenuated the Wnt/β-catenin signaling activity induced by Wnt3a alone or in combination with Wnt3a and R-spondin 2. These results indicate that LGR6 promotes myogenic differentiation and that ATRA is required for the transient expression of LGR6 during differentiation. Furthermore, LGR6 expression appeared to be downregulated by the ubiquitin–proteasome system involving ZNRF3.

## 1. Introduction

The skeletal muscle is composed of contractile and multinucleated muscle cells (muscle fibers) with remarkable plasticity. Exercise induces hypertrophy, and disuse induces muscle fiber atrophy. Muscle fibers are formed during developmental myogenesis or postnatal regeneration through highly orchestrated differentiation processes [1]. Therefore, identifying the factors that regulate myogenic differentiation and obtaining information on their expression mechanisms are important for understanding the complicated myogenic differentiation processes.

The myogenic precursor satellite cells differentiate into myoblasts. Subsequently, myoblasts differentiate and then undergo fusion, which involves a two-phase process, to form multinucleated myotubes [2]. In the first phase, myoblasts fuse with each other to form initial multinucleated cells (primary fusion), and in the second phase, myoblasts fuse with immature myotubes to form mature myotubes (secondary fusion). During differentiation, myotubes express myotube-specific structural proteins, such as myosin heavy chain (MyHC). Myogenic regulatory factors, such as MyoD, play a critical role in myoblast determination and myogenin functions in a genetic pathway downstream of MyoD [3]. Myomaker is essential for the hemifusion of the myoblast plasma membranes, and myomerger induces the fusion of non-fusogenic cells [4]. Furthermore, myogenic differentiation is regulated by various signaling molecules, such as vitamin A and Wnt ligands [5,6,7].

The vitamin A metabolite all-*trans* retinoic acid (ATRA) promotes myogenic differentiation in skeletal muscles [5,7]. ATRA acts as an agonist of the retinoic acid receptor (RAR), a nuclear receptor that functions as a transcription factor. RAR has three subtypes, α, β, and γ, which are encoded by different loci. The RAR forms a heterodimer with the retinoid X receptor and binds to the retinoic acid response element present in the promoter region of target genes in the nucleus. The binding of ATRA to RAR activates its heterodimer and induces the transcription of target genes. However, there are few reports on the target genes involved in promoting myogenic differentiation by ATRA.

Wnt signaling is an important pathway in embryonic muscle development and postnatal myogenesis [8]. The secreted Wnt ligands initiate signal transduction by binding to a heterodimeric receptor complex comprising a frizzled receptor (Fzd) and either low-density lipoprotein receptor-related protein 5 or low-density lipoprotein receptor-related protein 6, leading to the stabilization of the key effector β-catenin for the activation of signal transduction [9]. The activation of β-catenin-dependent Wnt signaling triggers the transcription of genes through the binding of a complex of β-catenin and TCF transcription factor to a specific promoter element, thus leading to the transition from cell proliferation to myogenic differentiation [10]. The four proteins in the R-spondin (RSPO) family, including RSPO2, enhance the Wnt/β-catenin signaling in the presence of Wnt ligands [11,12].

Leucine-rich repeat-containing G-protein-coupled receptors (LGR) 4, LGR5, and LGR6 are G-protein coupled receptors that belong to the rhodopsin-like seven-transmembrane receptor superfamily and are classified into group B of the LGR family [13]. These three LGRs act as receptors for the four RSPO proteins [14]. In the absence of RSPO proteins, the E3 ubiquitin ligase zinc and ring finger 3 (ZNRF3) and its homologous ring finger 43 (RNF43) ubiquitinate Fzd, resulting in a decrease in Wnt/β-catenin signaling. In contrast, in the presence of RSPO proteins, the interaction between RPSOs and ZNRF3 induces an association between ZNRF3 and LGR4, resulting in the activation of Wnt/β-catenin signaling due to the stabilization of Fzd [15,16]. The depletion of LGR4 represses myogenic differentiation and inhibits the RSPO2-mediated activation, but not the Wnt3a-mediated activation, of Wnt/β-catenin signaling [17]. LGR5-positive muscle progenitor cells contribute to muscle regeneration following injury [18]. RNA sequencing has shown that LGR6 is a candidate ATRA-responsive gene in myotubes [19]. However, the role of LGR6 in skeletal muscles remains unclear. On the other hand, recent studies have reported that *Lgr6* is expressed during the osteogenic differentiation of mesenchymal stem cells [20] and plays a critical role in bone homeostasis and regeneration [21,22]. RSPO2 stimulates Wnt/β-catenin signaling through LGR6 in the mouse mesenchymal stem cell line C3H10T1/2. Mesenchymal stem cells can differentiate into several lineages of mesenchymal tissues, including bone, cartilage, and muscle [23]. These results have led us to test whether LGR6 regulates myogenic differentiation in skeletal muscles.

In this study, we found that LGR6 was transiently expressed during myogenic differentiation and promoted myoblast differentiation and that LGR6 expression required ATRA. Finally, we analyzed the involvement of the ubiquitin–proteasome system in the regulation of LGR6 expression.

## 2. Results

### 2.1. LGR6 Is Involved in the Promotion of Myogenic Differentiation

We assessed the expression patterns of the mRNAs encoding LGR6 and myogenic regulatory factors during myogenic differentiation. The expression of *Lgr6* mRNA decreased 1 day after the induction of differentiation and remained low until 5 days after the induction of differentiation (Figure 1A). Because the expression of *Lgr6* mRNA was highest immediately after the induction of differentiation, we further evaluated the expression pattern of *Lgr6* mRNA up to 1 day after the induction of differentiation. The expression of *Lgr6* mRNA transiently increased 3 h after the induction of differentiation, followed by a decrease to basal levels 12 h after the induction of differentiation (Figure 1B). The expression of the mRNAs encoding myomaker, myomerger, and myogenin remained unchanged for up to 12 h but increased 24 h after the induction of differentiation. The expression of *myoD* mRNA decreased 12 h after the induction of differentiation. Furthermore, we assessed the expression pattern of the LGR6 protein during myoblast differentiation. In this experiment, a commercial anti-LGR6 antibody detected two bands (Figure 1C). The upper band was always detected during differentiation, whereas the lower band transiently increased 3 h after the induction of differentiation, followed by a decrease to basal levels 12 h after the differentiation induction (Figure 1D). The knockdown of LGR6 abolished only the lower band 3 h after the induction of differentiation, indicating that the lower band was specific to LGR6 (Appendix A). The expression of myomerger and myogenin increased with myogenic differentiation (Figure 1C,D). Next, to assess the role of LGR6 in myogenic differentiation, myoblasts were transfected with control siRNA or *Lgr6* siRNA and cultured in a differentiation medium for 3 days. The knockdown of *Lgr6* using siLGR6#1 and siLGR6#2 decreased *Lgr6* mRNA expression at 3 and 6 h after the induction of differentiation (Appendix A). *Lgr6* mRNA expression may have increased by 3 h after the induction of differentiation. Therefore, because it is important that *Lgr6* mRNA be knocked down upon the induction of differentiation, in the present study we evaluated whether *Lgr6* expression was knocked down at 0 h after the induction of differentiation. The knockdown of *Lgr6* using siLGR6#1 and siLGR6#2 resulted in decreases of approximately 70% and 60% in *Lgr6* mRNA expression, respectively (Figure 1E). Immunostaining for MyHC was performed, and the nuclei were counterstained with DAPI (Figure 1F). LGR6 depletion decreased the differentiation and fusion indices, increased the percentage of MyHC-positive cells with one or two nuclei, and decreased the percentage of MyHC-positive cells with three or more nuclei (Figure 1G). Moreover, the loss of LGR6 decreased the expression levels of *myogenin*, *myomaker*, and *myomerger* mRNAs but not of *myoD* mRNA (Figure 1H). LGR6 depletion decreased the expression of myomerger and myogenin at the protein level (Appendix A). These results indicate that a transient increase in LGR6 expression during differentiation precedes increased expression of factors involved in myoblast differentiation and fusion and that LGR6 promotes myogenic differentiation.

### 2.2. Transient Expression of LGR6 Appears to Be Required for the Promotion of Myogenic Differentiation

To study the effects of exogenous LGR6 expression on myogenic differentiation, myoblasts were transfected with the LGR6 expression vector, followed by the induction of differentiation. The expression of exogenous *Lgr6* mRNA for 3 h after the induction of differentiation led to the upregulation of the expression of *myoD*, *myogenin*, *myomaker*, and *myomerger* mRNAs (Figure 2A). In contrast, their expression significantly decreased, except for *myoD*, when exogenous *Lgr6* mRNA was sustainably expressed for 24 h after the induction of differentiation (Figure 2B). These results suggest that transient LGR6 expression promotes myogenic differentiation.

### 2.3. ATRA Is Required for Lgr6 mRNA Expression during Myogenic Differentiation

We investigated whether *Lgr6* was expressed as an ATRA-responsive gene during myogenic differentiation. Myoblasts were cultured in stripped differentiation medium in the presence or absence of ATRA. In the presence of ATRA, *Lgr6* mRNA expression transiently increased 3 h after the induction of differentiation, followed by a decrease to basal levels 12 h after the differentiation induction (Figure 3A). In the absence of ATRA, the expression of *Lgr6* mRNA remained unchanged after 12 h of differentiation. ATRA increased *Lgr6* mRNA expression, whereas the RAR-specific pan-antagonist AGN193109 completely inhibited this increase (Figure 3B). The transcriptional activity of RAR was measured in the presence of various concentrations of the RARα agonist AM580 (0.3–10 nM) and the RARγ agonist BMS961 (10–1000 nM). AM580 and BMS961 increased the transcriptional activity of RAR in a concentration-dependent manner (Appendix A). Similarly to ATRA, AM580, and BM961 increased *Lgr6* mRNA expression at concentrations of 1 nM and 100 nM, respectively, and co-treatment with the two agonists further increased its expression (Figure 3C). Next, the transcriptional activity of RAR was measured in the presence of ATAR (10 nM) and various concentrations of the RARα antagonist Ro41-5253 (0.1–10 µM) or the RARγ antagonist LY2955303 (10 and 50 nM). Ro41-5253 and LY2955303 inhibited the RAR transcriptional activity in a concentration-dependent manner (Appendix A, respectively). ATRA-induced upregulation of the *Lgr6* mRNA was suppressed to the same extent by Ro41-5253 or LY2955303 at concentrations of 300 nM and 100 nM, respectively (Figure 3D). Co-treatment with the two antagonists further suppressed its expression compared to the treatment with Ro41-5253 alone, but not with LY2955303 alone (*p* = 0.13). To assess the effects of RARα agonist and RARγ agonist on myogenic differentiation, myoblasts were cultured in stripped differentiation medium in the presence or absence of AM580 or BMS961. MyHC was immunostained, and the nuclei were counterstained with DAPI (Figure 3E). AM580 and BMS961 increased the differentiation and fusion indices (Figure 3F). In addition, to examine the role of ATRA in LGR6-mediated myogenic differentiation, myoblasts were transfected with control siRNA (siControl) or LGR6 siRNA (siLGR6#1) and differentiated for 3 days. The knockdown of *Lgr6* resulted in a decrease of approximately 64% in *Lgr6* mRNA expression (Figure 3G). MyHC was immunostained, and the nuclei were counterstained with DAPI (Figure 3H). The depletion of LGR6 reduced the differentiation and fusion indices in the presence or absence of ATRA but did not completely suppress the ATRA-induced increases in these indices (Figure 3I). These results indicate that ATRA is required for *Lgr6* mRNA expression during myogenic differentiation and suggest that RARα and RARγ are involved in ATRA-responsive *Lgr6* mRNA expression.

### 2.4. LGR6 Expression Is Downregulated by the Ubiquitin–Proteasome System

To assess whether LGR6 expression was regulated at the protein level, myoblasts were transfected with a human or mouse LGR6 expression vector and differentiated in the presence or absence of MG132. The expression of both exogenous LGR6 was increased by MG132 (Figure 4A). To determine whether LGR6 was associated with ubiquitinated conjugates, myoblasts were transfected with a human LGR6 expression vector and cultured in the presence or absence of MG132. Incubation of the cell lysates with Ni-Sepharose resin resulted in the pull-down of ubiquitinated conjugates with LGR6 (Figure 4B). Next, we investigated whether ZNRF3 was involved in the LGR6 expression. Myoblasts were transfected with a human LGR6 expression vector, followed by transfection with control siRNA or ZNRF3 siRNAs. The knockdown of *Znrf3* using two ZNRF3 siRNAs (siZNRF3#1 and siZNRF3#2) reduced *Znrf3* mRNA expression by approximately 62% and 61%, respectively (Figure 4C). A Western blot analysis revealed that loss of ZNRF3 increased LGR6 expression (Figure 4D). Furthermore, we assessed whether LGR6 was degraded by ubiquitination. Lysine residues in the intracellular regions deduced from the seven transmembrane domains of LGR6 were located at positions 598, 679, and 861 in human LGR6 and at positions 598, 679, 861, 924, 933, and 939 in mouse LGR6 [13,24]. Figure 4E shows the three common lysine residues in human and mouse LGR6. We generated expression vectors with three single mutations (K598R, K679R, and K861R), in which each lysine residue was replaced by an arginine residue, and a mutant carrying a triple mutation (K598/679/861R), in which all three lysine residues were replaced by arginine residues. None of these mutations increased LGR6 expression in the absence of MG132, whereas MG132 increased LGR6 expression (Figure 4F). These results suggest that LGR6 itself is not ubiquitinated, but LGR6 is degraded by the ubiquitin–proteasome system in a ZNRF3-dependent manner.

### 2.5. LGR6 Activates Wnt/β-Catenin Signaling

We assessed the effects of RSPO2 and/or Wnt3a on Wnt/β-catenin signaling in C2C12 cells. The Wnt/β-catenin signaling pathway was examined based on TCF-driven luciferase activity. RSPO2 did not affect TCF activity in the concentration range of 10–200 ng/mL (Figure 5A). In contrast, RSPO2 increased TCF activity in the presence of Wnt3a in a dose-dependent manner (Figure 5B). To investigate whether LGR6 was involved in Wnt/β-catenin signaling during myogenic differentiation, myoblasts were transfected with control siRNA or LGR6 siRNA and differentiated. The knockdown of *Lgr6* (siLGR6#1 and siLGR6#2) resulted in decreases of approximately 70% and 50% in *Lgr6* mRNA expression, respectively (Figure 5C). Subsequently, LGR6-depleted myoblasts were differentiated in the presence or absence of RSPO2 and/or Wnt3a, and Wnt/β-catenin signaling was assessed. The attenuation of LGR6 repressed the Wnt3a-induced and Wnt3a/RSPO2-induced TCF activity (Figure 5D). These results suggest that LGR6 activates Wnt/β-catenin signaling during myogenic differentiation.

## 3. Discussion

The identification of factors that promote myogenic differentiation is important for understanding the regulatory mechanisms underlying the complicated process of myogenic differentiation. ATRA promotes myogenic differentiation, and LGR6 is a candidate ATRA-responsive gene in C2C12 myotubes [19]. In this study, we analyzed the roles of LGR6 in myogenic differentiation.

Myoblast fusion consists of two phases: primary fusion and secondary fusion [2]. The transient increase in *Lgr6* mRNA expression preceded the upregulation of the *myomaker*, *myomerger*, and *myogenin* mRNAs, whereas the loss of *Lgr6* downregulated the expression of these mRNAs. At the protein level, LGR6 expression transiently increased during differentiation, preceding the increased expression of myogenin and myomerger, and loss of LGR6 suppressed their expression. In addition, the depletion of *Lgr6* reduced the differentiation and fusion indices, increased the percentage of MyHC-positive cells with one or two nuclei, and decreased the percentage of MyHC-positive cells with three or more nuclei. These results indicate that the loss of *Lgr6* suppresses the fusion between myoblasts (primary fusion). LGR4, similarly to LGR6, regulates Wnt/β-catenin signaling, and the knockdown of *Lgr4* increases the percentage of MyHC-positive cells with one nucleus and decreases the percentage of MyHC-positive cells with two to five nuclei, indicating that LGR4 suppresses the fusion between myoblasts [17]. Recently, other signaling factors regulating myoblast fusion have been identified. Dock1 promotes myoblast fusion through the activation of Rac1 [25]. The depletion of POFUT1 negatively affects secondary myogenic fusion by repressing NFATc2/IL4 signaling [26]. The inhibition of ERK signaling induces CaMKII-dependent fusion of myoblasts with early myotubes [27]. These results indicate that different signaling pathways, including Wnt/β-catenin signaling, regulate myoblast fusion.

Myomaker and myomerger are the target genes of Wnt/β-catenin signaling [28,29]. Wnt/β-catenin signaling increases the expression of *myoD* by the binding of β-catenin to the distal enhancer of the mouse *myoD* gene [30]. Myogenin upregulates the *myomaker* mRNA by binding to the *myomaker* promoter, thereby promoting myoblast differentiation and fusion [31]. The expression of myomaker and myomerger is low in proliferating myoblasts, but it increases after differentiation and is maintained during differentiation and myoblast fusion [5,32]. The depletion of myomaker or myomerger decreases the fusion index, but it has no effect on the differentiation index [32,33]. Myomaker and myomerger contribute to membrane hemifusion and the formation of fusion pores, respectively, and are involved in the fusion of myoblasts [34]. These results indicate that LGR6 functions as a factor that promotes myogenic differentiation, possibly before myomaker and myomerger act on myoblast fusion in the first phase.

A gain-of-function approach involving the overexpression of LGR6 up to 3 and 24 h after the induction of differentiation yielded the opposite results regarding the expression of the *myogenin*, *myomaker*, and *myomerger* mRNAs. Under physiological conditions, LGR6 expression was high 3 h after the induction of differentiation. Reflecting this physiological condition, when LGR6 was overexpressed up to 3 h after the induction of differentiation, the expression of *myoD*, *myogenin*, *myomaker*, and *myomerger* mRNAs was increased. In contrast, under physiological conditions, LGR6 expression returned to basal levels by 12 h after the induction of differentiation. When LGR6 was overexpressed up to 12 h after the induction of differentiation, which did not reflect this physiological condition, the expression of *myogenin*, *myomaker*, and *myomerger* mRNAs was reduced. These results suggested that sustained expression of LGR6 during differentiation negatively regulates myogenic differentiation. Therefore, LGR6 may need to be expressed transiently during myogenic differentiation. These findings indicate the importance of transient LGR6 expression in the promotion of myogenic differentiation and prompted us to investigate the regulation of LGR6 expression at the protein level. The degradation of intracellular proteins is mainly regulated by the ubiquitin–proteasome and autophagy systems [35]. The proteasome inhibitor used upregulated the LGR6 protein, and the ubiquitin-conjugated proteins were pulled down by LGR6, suggesting that LGR6 expression is downregulated by the ubiquitin–proteasome system. LGR4 forms a complex with ZNRF3 and RNF43, even in the absence of RSPO proteins [15], whereas LGR5 does not form a complex with ZNRF3 or RNF43 in the presence or absence of RSPO proteins [36]. In this study, the loss of ZNRF3 upregulated LGR6 expression in the absence of RSPO proteins. Thus, LGR6 expression at the protein level was regulated by ZNRF3, which was similar to the observation for LGR4 expression but unlike the observation for LGR5 expression. In addition, human LGR6 is deduced to have three lysine residues in its intracellular region [13,24]. Mutations that replace lysine residues with arginine residues had no effects on the LGR6 expression. These results suggest that LGR6 expression is regulated by the ubiquitin–proteasome system involving ZNRF3 but not by the ubiquitination of LGR6 itself.

LGR6 is expressed in the skin, small intestine, and bone marrow mesenchymal stem cells [37,38,39]. However, there are no reports on the regulatory mechanisms of LGR6 expression. Similarly to ATAR, an RARα agonist and an RARγ agonist increased *Lgr6* mRNA expression in C2C12 myoblasts, and an RAR pan-antagonist inhibited the ATAR-induced increase in *Lgr6* mRNA expression. The IC_50_ values of Ro41-5253 for RAR transcriptional activity are 60 nM (RARα), 2400 nM (RARβ), and 3300 nM (RARγ) [40]. The K_i_ values of LY2955303 for binding to RAR are >1700 nM (RARα), >2980 nM (RARβ), and 1.1 nM (RARγ) [41]. These results indicate that Ro41-5253 and LY2955303 function as specific inhibitors against RARα and RARγ, respectively, at appropriate concentrations but inhibit the activity of other RAR subtypes at excessive concentrations. Recently, these antagonists have been used to block retinoic acid signaling. ATRA at a concentration of 10 nM increases SOCS3 expression in bone marrow-derived cells, and Ro41-5253 at a concentration of 1 µM suppresses its increase [42]. LY2955303 at a concentration of 1 µM inhibited the expression of totipotency marker genes in totipotential stem cells, in which retinoic acid-signaling plays an important role during early development [43,44]. In the present study, each antagonist was used at a concentration that inhibited ATRA-increased *Lgr6* mRNA expression by approximately 40% (300 nM Ro41-5253 and 100 nM LY2955303). Co-treatment with both antagonists suppressed *Lgr6* expression to a greater extent than the treatment with Ro41-5253 alone. Unfortunately, co-treatment with both antagonists did not significantly suppress *Lgr6* expression more than the treatment with LY2955303 alone (*p* = 0.13). Treatment with LY2955303 at lower concentrations or at concentrations that do not affect other RARs may accentuate the combined effects of the two antagonists. Furthermore, a loss-of-function approach revealed that ATRA was required for the transient expression of *Lgr6* mRNA during myogenic differentiation. The exogenous expression of RARα promotes myogenic differentiation and increases MyHC expression in C2C12 cells [5]. The oral administration of an RARγ-selective agonist to mice accelerates recovery from muscle damage; moreover, its repair is remarkably delayed in RARγ knockout mice [45]. Thus, ATRA promotes myogenic differentiation through RARα and RARγ, and RAR-mediated ATRA-responsive genes are involved in myogenic differentiation. Taken together, our results suggest that LGR6 contributes to the promotion of myogenic differentiation as an ATRA-responsive gene via RARα and RARγ.

Wnt/β-catenin signaling regulates multiple steps of myogenic differentiation [46]. During differentiation, a blockage of Wnt/β-catenin signaling inhibits myoblast fusion, suggesting that Wnt/β-catenin signaling plays a critical role in differentiation before and during myoblast fusion. The knockdown of *Lgr6* repressed Wnt3a-activated and Wnt3a/RSPO2-activated Wnt/β-catenin signaling and increased the proportion of MyHC-positive cells with one or two nuclei. Therefore, LGR6 appeared to promote myogenic differentiation by activating Wnt/β-catenin signaling before myoblast fusion. In contrast, *Lgr4* knockdown represses RSPO2-activated Wnt/β-catenin signaling and has no effect on Wnt3a-activated Wnt/β-catenin signaling [17]. In the presence of RSPOs, these proteins bind to LGR4 and then to ZNRF3 or its homologous RNF43 to form a complex, resulting in the prevention of Fzd receptor degradation and the activation of Wnt/β-catenin signaling [15]. In the absence of RSPOs, ZNRF3 and RNF43 ubiquitinate Fzd and promote its degradation, resulting in decreased Wnt/β-catenin signaling. These results suggest that LGR6 activates Wnt/β-catenin signaling via a molecular mechanism that is different from that observed for LGR4.

In conclusion, our study demonstrates that LGR6 promotes myogenic differentiation, perhaps by activating Wnt/β-catenin signaling before the first phase of myoblast fusion, and that ATRA is required for LGR6 expression. Furthermore, LGR6 expression appeared to be downregulated by the ZNRF3-mediated ubiquitin-proteasome system. These results highlight the importance of LGR6 in improving skeletal muscle regeneration.

## 4. Materials and Methods

### 4.1. Cell Culture

Murine C2C12 myoblasts (European Collection of Authenticated Cell Cultures, Salisbury, UK) were cultured in Dulbecco’s modified Eagle’s medium supplemented with 10% fetal bovine serum and antibiotics (growth medium) and differentiated into myotubes in Dulbecco’s modified Eagle’s medium supplemented with 2% horse serum and antibiotics (differentiation medium), as described previously [19]. To determine the effects of vitamin A on LGR6 expression, myoblasts were cultured in a growth medium supplemented with 10% dextran-coated charcoal-stripped fetal bovine serum (stripped growth medium), followed by culturing in a differentiation medium supplemented with 2% dextran-coated charcoal-stripped horse serum (stripped differentiation medium) in the presence or absence of an RAR agonist (10 nM ATRA, 3 or 10 nM AM580 (Tokyo Chemical Industry Co., Ltd., Tokyo, Japan), or 100 or 300 nM BMS961 (Tocris Bioscience, Bristol, UK)). To assess the involvement of ATRA and RAR in *Lgr6* expression, myoblasts were cultured for 6 h in a stripped growth medium with an RAR agonist (10 nM ATRA, 1 nM AM580, or 100 nM BMS961), with ATRA in the presence or absence of an RAR pan-antagonist (10 nM ATRA and 100 nM AGN193109 (Toronto Research Chemicals Inc., Toronto, Canada)), or with ATRA in the presence or absence of an RAR-subtype-specific antagonist (10 nM ATRA and 300 nM Ro41-5253 (Focus Biomolecules, Plymouth Meeting, PA, USA) or 100 nM LY2955303 (Cayman, Ann Arbor, MI, USA)).

### 4.2. Plasmids

Murine *Lgr6* cDNA was amplified by nested PCR. The C-terminal Myc- and His-tagged murine LGR6 expression vector (termed pcDNA3.1-mLGR6-myc-his) was then constructed. Human LGR6 cDNA was amplified by PCR using pFN21A-Lgr6 (clone name: pFN21AE9442; Kazusa Genome Technologies Inc., Chiba, Japan) and subcloned into the pcDNA3.1-myc-his vector to construct the C-terminal Myc- and His-tagged human LGR6 expression vector (termed pcDNA3.1-hLGR6-myc-his). To assess the ubiquitination of LGR6, the following four types of LGR6 expression vectors were synthesized via site-directed mutagenesis of pcDNA3.1-hLGR6-myc-his, which is a single mutation with a lysine-to-arginine substitution at amino acid positions 598, 679, or 861, and a triple mutation encompassing these three mutations.

### 4.3. Immunofluorescence Assay

Cells were fixed in 4% paraformaldehyde in PBS and permeabilized with 0.2% Triton X-100 in PBS, as described previously [47]. The cells were blocked and incubated with a mouse monoclonal anti-MyHC antibody (clone MF-20; Developmental Studies Hybridoma Bank, Iowa City, IA, USA), followed by incubation with Alexa 488-conjugated anti-mouse IgG. Cells were then counterstained with 4′,6-diamidino-2-phenylindole dihydrochloride (DAPI). Fluorescent images were analyzed using a BIOREVO BZ-800 fluorescence microscope (Keyence, Osaka, Japan). To assess the differentiation of myoblasts into myotubes, a differentiation index was calculated as the proportion of MyHC-positive cells with one or more nuclei. To assess myotube formation, the fusion index was calculated as the proportion of MyHC-positive cells with two or more nuclei.

### 4.4. siRNA-Mediated Knockdown

The sequences of murine *Lgr6* siRNA and murine *Znrf3* siRNA (Sigma-Aldrich; St. Louis, MO, USA) were as follows: LGR6#1, 5′-GGACUUUCAUCCAGAGGAAdTdT-3′; LGR6#2, 5′-GUCUAAACUGCAUACGCUAdTdT-3′; ZNRF3#1, 5′-CCUGUAUGGUCUUCACUCAdTdT-3′; and ZNRF3#2, 5′-CUCUUAAGAGGCCAGUGGUdTdT-3′. The control siRNA (MISSION siRNA Universal Negative Control) was purchased from Sigma-Aldrich. To assess LGR6 expression, C2C12 myoblasts were transiently transfected with control siRNA or *Lgr6* siRNA using Lipofectamine RNAiMAX reagent (Thermo Fisher Scientific, Inc., Waltham, MA, USA). Cells were cultured in a growth medium or stripped growth medium for 36 h, followed by culturing in a differentiation medium or stripped differentiation medium for 24 h (to assess myogenic regulatory factors) or 72 h (to calculate differentiation and fusion indices). For the assessment of ZNRF3, myoblasts were transfected with pcDNA3.1-myc-his (mock) or pcDNA3.1-hLGR6-myc-his in a growth medium for 24 h, followed by transfection with *Znrf3* siRNA for 48 h.

### 4.5. Transfection of Plasmids

For the assessment of myogenic regulatory factors, C2C12 myoblasts were transfected with pcDNA3.1-myc-his (mock) or pcDNA3.1-hLGR6-myc-his using the viofectin transfection reagent (Viogene; New Taipei City, Taiwan) in a growth medium for 24 h, followed by culturing in differentiation medium for 3 or 24 h. For the assessment of exogenous LGR6 expression, myoblasts were transfected with mock vector or LGR6 expression vectors for 24 h, followed by culturing in a growth medium in the presence or absence of the proteasome inhibitor MG132 (5 µM) for 24 h.

### 4.6. Quantitative PCR

Total RNA was extracted from C2C12 cells and reverse-transcribed using random hexamers. The resulting cDNA was subjected to quantitative real-time PCR (qPCR) using SYBR Green (TP-800; Takara Bio Inc., Shiga, Japan). The data were normalized to *18S rRNA* expression levels. The following specific primers were used: *Lgr6* forward, 5′-CCTGATGCACCTGAAGCTCAAAG-3′ and reverse, 5′-GTGGAGGCCAGAGAATGCC-3′; *MyoD* forward, 5′-TGGGATATGGAGCTTCTATCGC-3′ and reverse, 5′-GGTGAGTCGAAACACGGGTCAT-3′; *Myogenin* forward, 5′-CATCCAGTACATTGAGCGCCTA-3′ and reverse, 5′-GAGCAAATGATCTCCTGGGTTG-3′; *Myomaker* forward, 5′-ATCGCTACCAAGAGGCGTT-3′ and reverse, 5′-CACAGCACAGACAAACCAGG-3′; *Myomerger* forward, 5′-CAGGAGGGCAAGAAGTTCAG-3′ and reverse, 5′-ATGTCTTGGGAGCTCAGTCG-3′; and *18S rRNA* forward, 5′-GTAACCCGTTGAACCCCATT-3′ and reverse, 5′-GGCCTCACTAAACCATCCAA-3′. Ct values were converted into relative quantification data using the 2^−ΔΔCt^ method.

### 4.7. Western Blotting

For the determination of endogenous LGR6 expression, C2C12 myoblasts were cultured in a differentiation medium. Cells were then sonicated in lysis buffer (50 mM Tris-HCl, pH 7.5, 1% Triton X-100, 150 mM NaCl, 1% sodium deoxycholate, 0.1% SDS, 2 mM EDTA, 1 mM phenylmethylsulfonyl fluoride, 1 μg/mL aprotinin, 10 μg/mL leupeptin, 1 mM DTT, 50 mM sodium fluoride, 1 mM sodium orthovanadate, 10 mM sodium molybdate, and 10 mM sodium pyrophosphate) to prepare cell lysates. For the pull-down assay, the cell lysates were incubated with Ni-Sepharose 6 Fast Flow resin for 2 h at 4 °C, followed by washing with lysis buffer. Cell lysates (input) and resin-bound proteins were prepared. Proteins were subjected to SDS–PAGE, followed by Western blot analysis using the following primary antibodies: rabbit monoclonal anti-LGR6 (clone EPR6874; Abcam, Cambridge, UK), sheep polyclonal anit-myomerger (R&D Systems, Minneapolis, MN, USA), mouse monoclonal anti-myogenin (Developmental Studies Hybridoma Bank, Iowa city, IA, USA), mouse monoclonal anti-β-actin (clone 2D4H5; Proteintech Group, Inc., Rosemont, IL, USA), anti-Myc (clone 9B11; Cell Signaling Technology, Danvers, MA, USA), and rabbit polyclonal anti-ubiquitin (Cell Signaling Technology) antibodies. Horseradish-peroxidase-conjugated goat anti-mouse or anti-rabbit IgG was used as a secondary antibody. The blots were treated with the Immobilon Western Chemiluminescent HRP Substrate (Millipore, Billerica, MA, USA).

### 4.8. Reporter Assay

C2C12 myoblasts were transfected with the pTCF7wtluc reporter vector (RIKEN BioResource Research Center, Ibaraki, Japan) and the pRL-TK control reporter vector (Promega, Madison, WI, USA) for 24 h. To test the effects of Wnt3a, cells were transfected with either pcDNA3.2 or pcDNA3.2-Wnt3a-V5 (Addgene, Watertown, MA, USA) together with the reporter vectors, followed by transfection with 20 nM control siRNA or *Lgr6* siRNA for 36 h. The cells were then incubated in a growth medium or differentiation medium containing RSPO2 at a final concentration of 20 ng/mL for 24 h. Subsequently, the cells were lysed and firefly and *Renilla* luciferase activities were determined using the Dual-Luciferase Reporter Assay Kit and a GloMax 20/20 Luminometer (Promega). The transfection efficiency was normalized using pRL-TK. Data were expressed as relative light units (firefly luciferase activity divided by *Renilla* luciferase activity).

### 4.9. Statistics

Data are expressed as the mean ± S.D. and were analyzed using Student’s *t*-test, one-way ANOVA, or two-way ANOVA with Dunnett’s post hoc test or Tukey’s post hoc test. Statistical analyses were performed using the JMP statistical software, version 8.0.1 (SAS Institute, Cary, NC, USA). Differences were considered statistically significant at *p* < 0.05.

## Figures and Tables

**Figure 1 ijms-24-09035-f001:**
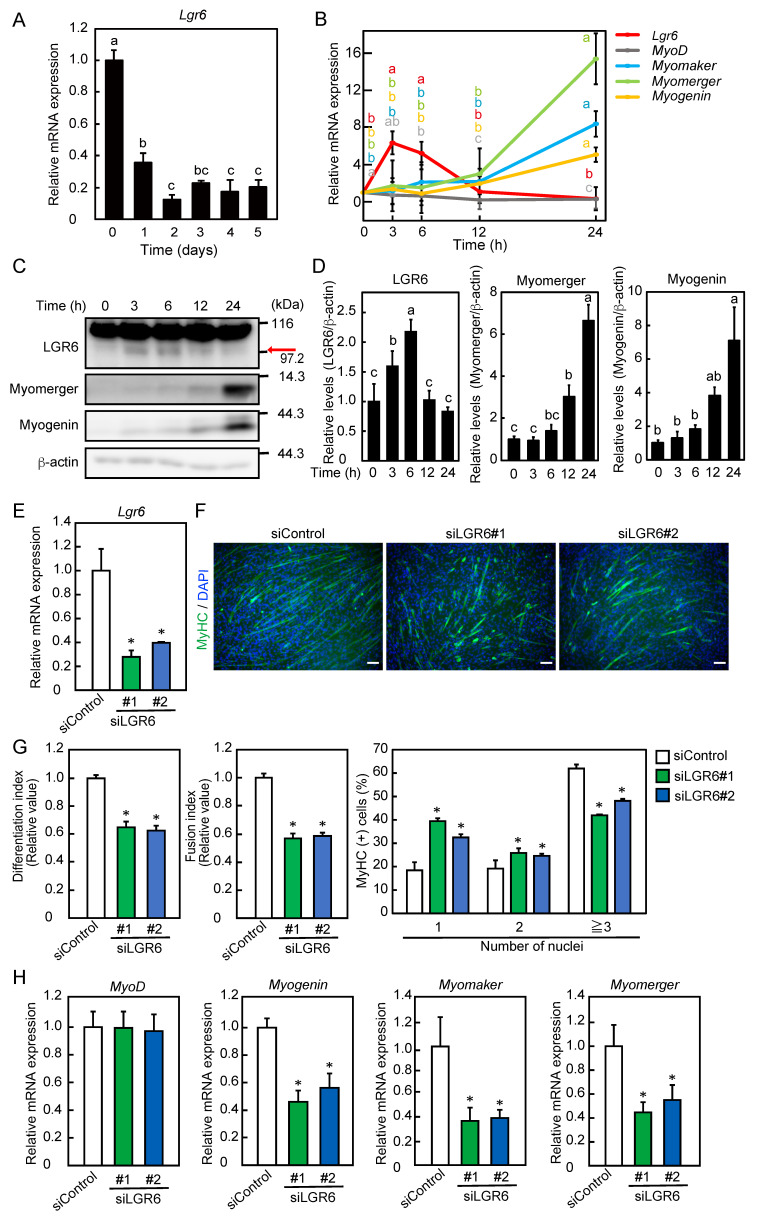
Expression pattern and role of LGR6 during myoblast differentiation. (**A**) C2C12 myoblasts were differentiated into myotubes for 5 days. The expression level of *Lgr6* mRNA was determined by qPCR. (**B**) Myoblasts were differentiated into myotubes for 24 h. The levels of mRNAs encoding LGR6 and myogenic regulatory factors were determined by qPCR. (**C**) Myoblasts were differentiated into myotubes. LGR6, myomerger, and myogenin expression was analyzed by Western blotting. The arrow indicates the lower band of LGR6. (D) LGR6, myomerger, and myogenin levels were normalized to β-actin levels. (**E**) Myoblasts were transfected with control siRNA (siControl) or *Lgr6* siRNA (siLGR6 #1 and siLGR6 #2), and cells were harvested 0 h after the induction of differentiation. The *Lgr6* mRNA levels were determined by qPCR. (**F**) Myoblasts were differentiated for 3 days after siRNA transfection. Fixed cells were fluorescently labeled using an anti-MyHC antibody and fluorescence-labeled secondary antibody (green). The nuclei were stained with DAPI (blue). Bars, 100 μm. (**G**) The differentiation and fusion indices were calculated. The percentage of MyHC-positive cells with one, two, or three more nuclei was determined. (**H**) Myoblasts were differentiated for 24 h after siRNA transfection, and cells were harvested. The levels of mRNAs for myogenic regulatory factors were determined by qPCR. (**A**,**B**,**D**) The results are presented as the mean ± SD (*n* = 3). Data were determined using one-way ANOVA and Tukey’s post hoc test. (**A**,**D**) Columns with different letters are significantly different at *p* < 0.05, whereas columns sharing the same letters are not significantly different. (**B**) Different letters with the same color on the lines indicate statistically significant differences (*p* < 0.05). (**E**,**G**,**H**) The results are presented as the mean ± SD (*n* = 3). Data were determined using one-way ANOVA and Dunnet’s post hoc test. * *p* < 0.05 vs. siControl.

**Figure 2 ijms-24-09035-f002:**
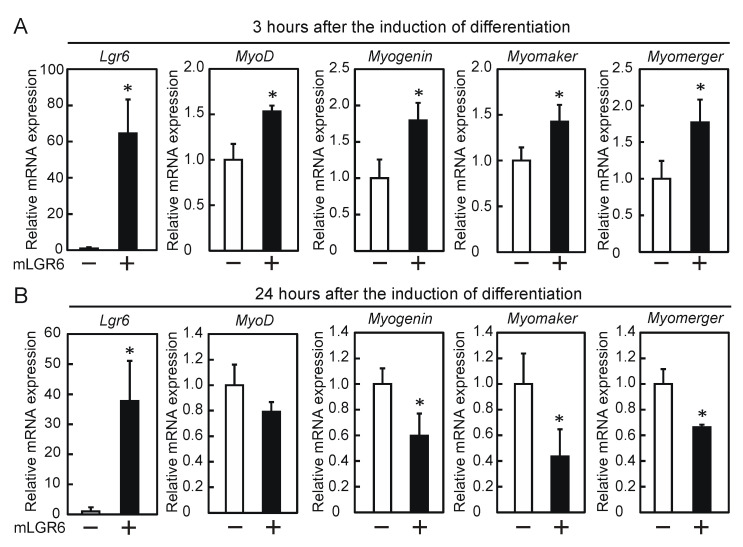
Effects of the overexpression of LGR6 on myogenic differentiation. Myoblasts were transfected with a mock vector or human LGR6 expression vector, followed by the induction of differentiation. The levels of mRNAs encoding LGR6 and myogenic regulatory factors were determined by qPCR. (**A**) Three hours after the induction of differentiation. (**B**) Twenty-four hours after the induction of differentiation. The results are presented as the mean ± SD (*n =* 3). Data were determined by Student’s *t*-test. * *p* < 0.05 vs. mock.

**Figure 3 ijms-24-09035-f003:**
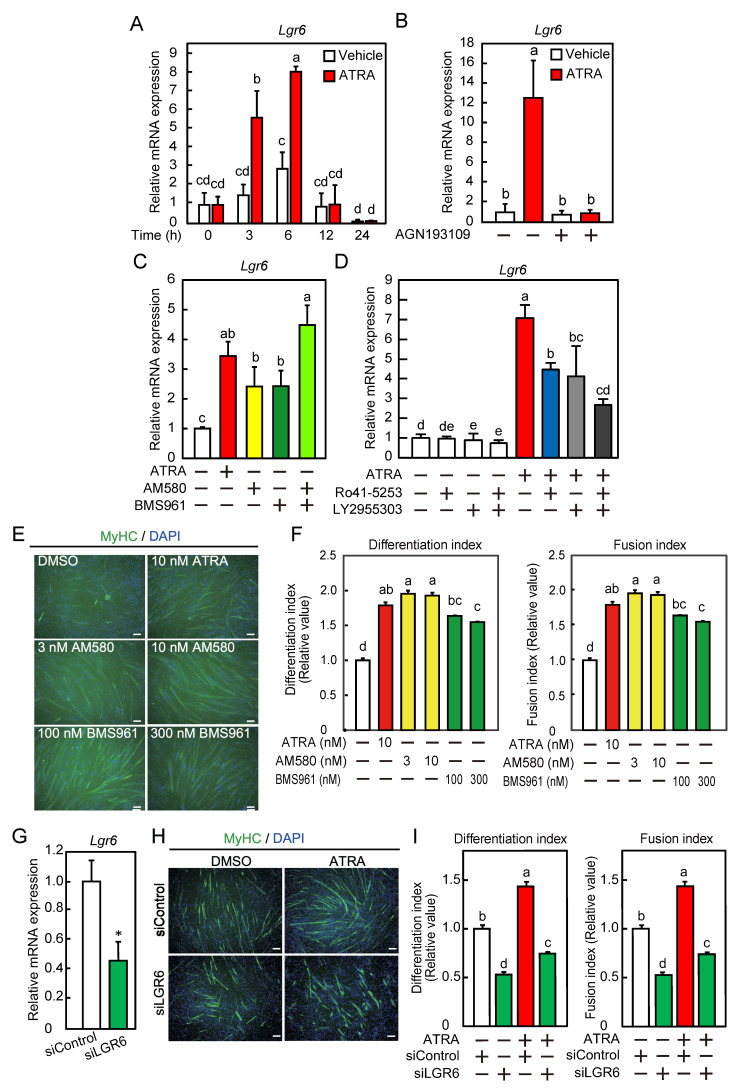
LGR6 expression during myoblast differentiation as an ATRA-responsive gene. (**A**) C2C12 myoblasts were differentiated in the presence or absence of ATRA. The *Lgr6* mRNA levels were determined by qPCR. (**B**) Myoblasts were cultured with ATRA in the presence of AGN193109. The *Lgr6* mRNA levels were determined by qPCR. (**C**) C2C12 myoblasts were cultured with ATRA, AM580, BMS961, AM580, and BMS961. The *Lgr6* mRNA levels were determined by qPCR. (**D**) Myoblasts were cultured with ATRA in the presence or absence of Ro41-5253 and/or LY2955303. The *Lgr6* mRNA levels were determined by qPCR. (**E**) Myoblasts were differentiated in the presence or absence of an RAR agonist. Fixed cells were immunofluorescently labeled using an anti-MyHC antibody (green), and the nuclei were stained with DAPI (blue). Bars, 100 μm. (**F**) The differentiation and fusion indices were calculated. (**G**) Myoblasts were transfected with control siRNA (siControl) or *Lgr6* siRNA (siLGR6#1), and cells were harvested immediately after the induction of differentiation. The *Lgr6* mRNA levels were determined by qPCR. (**H**) After siRNA transfection, myoblasts were differentiated in the presence or absence of ATRA for 3 days. Fixed cells were immunofluorescently labeled using an anti-MyHC antibody (green), and the nuclei were stained with DAPI (blue). Bars, 100 μm. (**I**) The differentiation and fusion indices were calculated. (**A**–**D**,**F**,**I**) The results are presented as the mean ± SD (*n* = 3). Data were determined using two-way ANOVA and Tukey’s post hoc test. Columns with different letters are significantly different at *p* < 0.05, whereas columns sharing the same letters are not significantly different. (**G**) The results are presented as the mean ± SD (*n* = 3). Data were determined using Student’s *t*-test. * *p* < 0.05 vs. siControl.

**Figure 4 ijms-24-09035-f004:**
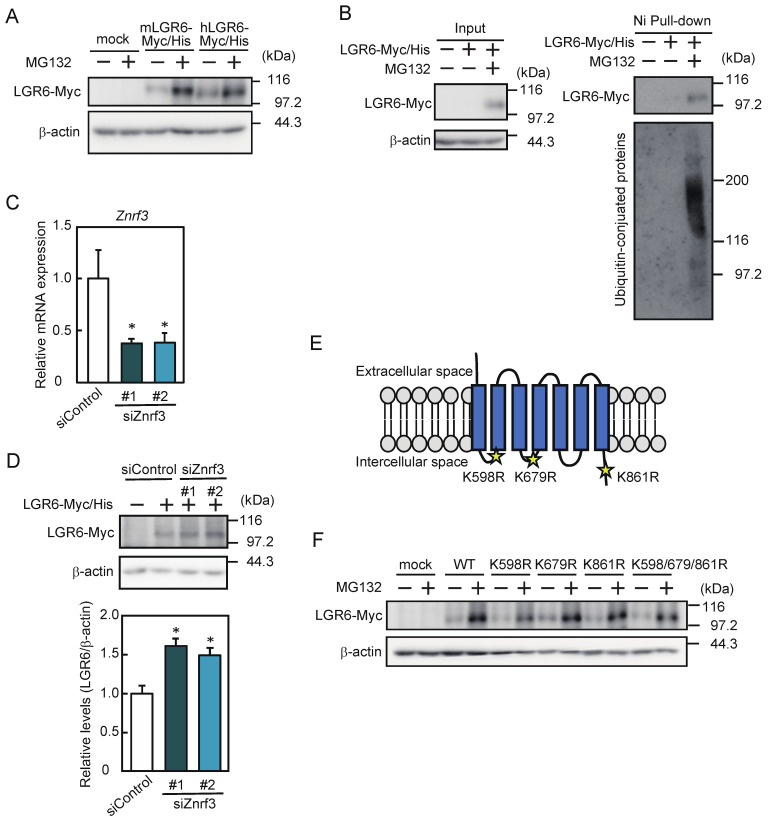
Involvement of the ubiquitin–proteasome system in LGR6 expression. (**A**) C2C12 myoblasts were transfected with a mock vector and murine and human LGR6 expression vectors, followed by culture in the presence or absence of MG132. LGR6 levels were analyzed by Western blotting. (**B**) Myoblasts were transfected with a mock vector and a human LGR6 expression vector, followed by further culture in the presence or absence of MG132. The LGR6 levels were analyzed by Western blotting. Myc- and His-tagged LGR6 (LGR6-Myc/His) was pulled down using Ni-Sepharose resin, and Ni-Sepharose-bound proteins were analyzed by Western blotting using anti-Myc and anti-ubiquitin antibodies. (**C**) Myoblasts were transfected with a human LGR6 expression vector, followed by further transfection with control siRNA or *Znrf3* siRNAs (siZNRF3#1 and siZNRF3#2). As a negative control for exogenous LGR6, myoblasts were transfected with a mock vector. The *Lgr6* mRNA levels were determined by qPCR. Data were determined by Student’s *t*-test. * *p* < 0.05 vs. siControl. (**D**) Exogenous LGR6 levels were analyzed by Western blotting and normalized to β-actin levels. The results are presented as the mean ± SD (*n* = 3). Data were determined using one-way ANOVA and Dunnet’s post hoc test. * *p* < 0.05 vs. siControl. (**E**) Deduced positions of lysine residues (yellow stars: K598, K679, and K861) in the intracellular region of human LGR6. (**F**) Myoblasts were transfected with a mock vector and wild-type or mutant-type human LGR6 expression vectors, followed by further culture in the presence or absence of MG132. LGR6 expression was analyzed by Western blotting using an anti-Myc antibody.

**Figure 5 ijms-24-09035-f005:**
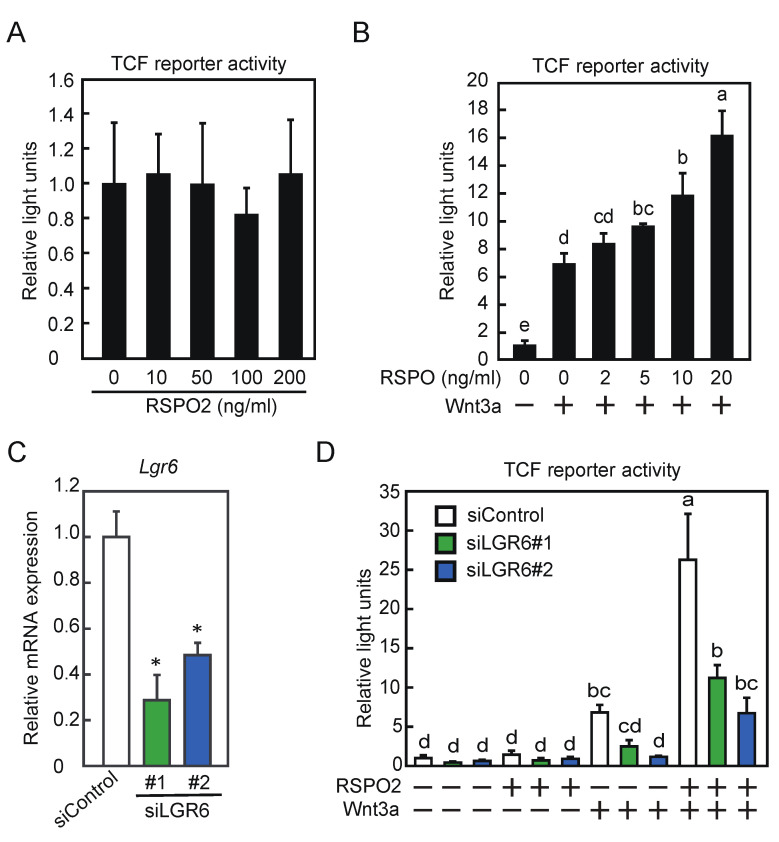
Involvement of LGR6 in Wnt/β-catenin signaling. (**A**) C2C12 myoblasts were transfected with a TCF reporter expression vector, followed by further incubation in the presence or absence of RSPO2. TCF activity was determined. (**B**) Myoblasts were transfected with a TCF reporter expression vector and a Wnt3a expression vector, followed by further incubation in the presence or absence of RSPO2. TCF activity was determined. (**C**) Myoblasts were transfected with a TCF reporter expression vector and a Wnt3a expression vector and further transfected with control siRNA (siControl) or *Lgr6* siRNAs (siLGR6 #1 and siLGR6 #2). Cells were harvested immediately after the induction of differentiation. The *Lgr6* mRNA levels were determined by qPCR. (**D**) After siRNA transfection, myoblast differentiation was induced in the presence or absence of RSPO2. TCF activity was determined. (**A**,**B**,**D**) The results are presented as the mean ± SD (*n* = 3). Data were determined using two-way ANOVA and Tukey’s post hoc test. Columns with different letters are significantly different at *p* < 0.05, whereas columns sharing the same letters are not significantly different. (**C**) The results are presented as the mean ± SD (*n* = 3). Data were determined using Student’s *t*-test. * *p* < 0.05 vs. siControl.

## Data Availability

The data that support the findings of this study are available from the corresponding authors upon reasonable request.

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
