# Peer review of "All-Trans Retinoic Acid-Responsive LGR6 Is Transiently Expressed during Myogenic Differentiation and Is Required for Myoblast Differentiation and Fusion"

_ijms, 2023, doi:10.3390/ijms24109035_

Round 1

Reviewer 1 Report

Reviewer’s Comments

The manuscript “All-trans Retinoic Acid-responsive LGR6 Functions as a Positive Regulator of Myogenic Differentiation” is a very interesting work. In this work, All-trans retinoic acid (ATRA) promotes the differentiation of myoblasts to myotubes. The leucine-rich repeat-containing G-protein-coupled receptor 6 (LGR6) is a candidate ATRA-responsive gene, but its role in skeletal muscle remains unclear. Here, we demonstrated that during the differentiation of murine C2C12 myoblasts to myotubes, Lgr6 mRNA expression is transiently increased before the increase in the expression of the mRNAs encoding myogenic regulatory factors, such as myogenin, myomaker, and myomerger. Loss of LGR6 decreased the differentiation and fusion indices. Exogenous expression of LGR6 up to 3 h and 24 h after the induction of differentiation increased and decreased the mRNA levels of the myogenin, myomaker, and myomerger, respectively. The Lgr6 mRNA was transiently expressed after myogenic differentiation in the presence of a retinoic acid receptor α (RARα) agonist and a RARγ agonist besides ATRA, but not in the absence of ATRA. Furthermore, a proteasome inhibitor or ZNRF3 knockdown increased exogenous LGR6 expression. Loss of LGR6 attenuated the Wnt/β-catenin signaling activity induced by Wnt3a alone or in combination with Wnt3a and R-spondin 2.  While I believe this topic is of great interest to our readers, I think it needs major revision before it is ready for publication. So, I recommend this manuscript for publication with major revisions.

1. In this manuscript, the authors did not explain the importance of LGR6 Functions in the introduction part. The authors should explain the importance of LGR6 Functions.

2) Title: The title of the manuscript is not impressive. It should be modified or rewritten it.

3) Correct the following statement “These results indicate that transient expression of LGR6 requires ATRA and promotes myogenic differentiation. Furthermore, LGR6 expression appeared to be regulated by the ubiquitin-proteasome system involving ZNRF3”.

4) Keywords: The LGR6 Functions is missing in the keywords. So, modify the keywords.

5) Introduction part is not impressive. The references cited are very old. So, Improve it with some latest literature such as 10.3390/molecules27217368, 10.3390/molecules27207129

6) The authors should explain the following statement with recent references, “The ATRA-induced upregulation of the Lgr6 mRNA was suppressed by the RARα antagonist Ro41-5253 or the RARγ antagonist LY2955303 (Figure 3D)”.

7) Add space between magnitude and unit. For example, in synthesis “21.96g” should be 21.96 g. Make the corrections throughout the manuscript regarding values and units.

8) The author should provide reason about this statement “In addition, depletion of Lgr6 reduced the differentiation and fusion indices and downregulated the myomaker, my-omerger, and myogenin mRNAs, indicating that loss of LGR6 delays myogenic differentiation”.

9. Comparison of the present results with other similar findings in the literature should be discussed in more detail. This is necessary in order to place this work together with other work in the field and to give more credibility to the present results.

10) Conclusion part is very long. Make it brief and improve by adding the results of your studies.

11) There are many grammatic mistakes. Improve the English grammar of the manuscript.

 Minor editing of English language required

Reviewer 2 Report

In this manuscript, Kitakaze and colleagues investigate functions of LGR6 in differentiation of C2C12 myoblasts to myotubes. They show that LGR6 expression increases transiently upon the induction of myogenic differentiation by switching culture conditions. They provide evidence that LGR6 regulates differentiation and fusion of C2C12 cells, LGR6 expression is regulated by retinoic acid signaling, LGR6 is degraded in a Znrf3-dependent and ubiquitin-independent manner, and LGR6 activates Wnt canonical signaling activity. Overall, the experiments are conducted appropriately, and the results look convincing. The manuscript is written succinctly and is easy to follow. It would have been more comprehensive if the authors investigated the effect of Wnt3a/RSPO2 on C2C12 differentiation and fusion and how it changes by LGR6 knockdown, but this might be too much to ask for this journal.

However, one key information that is missing in this manuscript is whether the expression of LGR6 stays low beyond 24 hours after the induction of C2C12 differentiation or it comes up again later during C2C12 differentiation process. The effect of LGR6 knockdown on C2C12 differentiation and fusion (tested 72 hours after switching media) might reflect LGR6 expression and function beyond the first 24 hours, rather than the mechanism proposed by the authors that LGR6 functions early in the differentiation process. This information needs to be provided before publication of this manuscript.

Minor comments

      While the authors concisely provide their method, it is hard to know when samples were collected in each experiment. It would be helpful to specify in the figure legends how many hours after induction these samples were collected. For example, when was LGR6 expression tested in Figure 1B? 24 hours after induction? 3-6 hours? 3 days, as shown in Figure 1C? This information is critical to know siRNA efficiency in this study. I think it is essential to test LGR6 siRNA efficiency 3-6 hours after induction, since the authors investigate LGR6 functions during this time period. Supplementary Figure S1 partially address this point, but LGR6 mRNA levels need to be also tested. 

      What are the alphabetical labels in some of the graphs (Fig 1A, 1G, 3A­–D, etc)? They are presumably statistical details, but the information cannot be found anywhere in the manuscript.

      Pictures in Figure 1C and 3F need to be bigger and have better resolution. Current images do not represent corresponding quantifications.

      What happens to MyHC expression when LGR6 is overexpressed? How do differentiation and fusion change? 

      While the authors investigate MyoD expression in some of the experiments, they never mention it. MyoD expression is increased by LGR6 overexpression in Figure 2A, what do the authors think about this?

      What happens to LGR6 mRNA expression when both RAR antagonists are added together? Would this be toxic for the cells?

Round 2

Reviewer 2 Report

The authors substantially improved their manuscript. However, it appears that Figure 1 has not been fully updated in their revised manuscript (I don't find new Figure 1A).  Please carefully update the manuscript before publication.

Regarding the alphabetical letters in some of the graphs, it would be clearer to state something like "columns with different letters are significantly different at p < 0.05, while columns sharing the same letters are not significantly different" as described in one of the MDPI papers mentioned in the authors' response (https://www.mdpi.com/2072-6643/15/9/2032). 
